# Short-Chain Fatty Acids Attenuate 5-Fluorouracil-Induced THP-1 Cell Inflammation through Inhibiting NF-κB/NLRP3 Signaling via Glycerolphospholipid and Sphingolipid Metabolism

**DOI:** 10.3390/molecules28020494

**Published:** 2023-01-04

**Authors:** Yanyan Zhang, Yue Xi, Changshui Yang, Weijuan Gong, Chengyin Wang, Liang Wu, Dongxu Wang

**Affiliations:** 1Testing Center, Yangzhou University, Yangzhou 225009, China; 2Medical Laboratory Department, Huai’an Second People’s Hospital, Huai’an 223022, China; 3School of Medicine, Yangzhou University, Yangzhou 225009, China; 4Department of Laboratory Medicine, School of Medicine, Jiangsu University, Zhenjiang 212013, China; 5School of Grain Science and Technology, Jiangsu University of Science and Technology, Zhenjiang 212100, China

**Keywords:** short-chain fatty acids, 5-fluorouracil, inflammation, NLRP3 inflammasome, metabolomics, sphingolipid metabolism

## Abstract

5-Fluorouracil (5-FU) is a common anti-tumor drug, but there is no effective treatment for its side effect, intestinal mucositis. The inflammatory reaction of macrophages in intestinal mucosa induced by 5-FU is an important cause of intestinal mucositis. In this study, we investigated the anti-inflammatory effects of the three important short-chain fatty acids (SCFAs), including sodium acetate (NaAc), sodium propionate (NaPc), and sodium butyrate (NaB), on human mononuclear macrophage-derived THP-1 cells induced by 5-FU. The expressions of intracellular ROS, pro-inflammatory/anti-inflammatory cytokines, as well as the nuclear factor-κB/NLR family and pyrin domain-containing protein 3 (NF-κB/NLRP3) signaling pathway proteins were determined. Furthermore, the cell metabolites were analyzed by untargeted metabolomics techniques. Our results revealed that the three SCFAs inhibited pro-inflammatory factor expressions, including IL-1β and IL-6, when treated with 5-FU (*p* < 0.05). The ROS expression and NF-κB activity of 5-FU-treated THP-1 cells were inhibited by the three SCFAs pre-incubated (*p* < 0.05). Moreover, NLRP3 knockdown abolished 5-FU-induced IL-1β expression (*p* < 0.05). Further experiments showed that the three SCFAs affected 20 kinds of metabolites that belong to amino acid and phosphatidylcholine metabolism in THP-1 cells. These significantly altered metabolites were involved in amino acid metabolism and glycerolphospholipid and sphingolipid metabolism. It is the first time that three important SCFAs (NaAc, NaPc, and NaB) were identified as inhibiting 5-FU-induced macrophage inflammation through inhibiting ROS/NF-κB/NLRP3 signaling pathways and regulating glycerolphospholipid and sphingolipid metabolism.

## 1. Introduction

Short-chain fatty acids (SCFAs) are important metabolites of intestinal flora, which are produced by the fermentation of indigestible dietary fiber and resistant starch by certain specific anaerobic bacteria in the gut [1,2]. The number of carbon atoms of SCFAs is less than six, and sodium acetate (NaAc), sodium propionate (NaPc), and sodium butyrate (NaB) together account for more than 90% of the total SCFAs, often in an ionic state in the intestinal tract [3]. SCFAs can promote the proliferation and differentiation of intestinal mucosa epithelial cells, prevent intestinal mucosa epithelial cells from atrophy, and maintain a normal mucosa barrier [4]. They also inhibit the production of pro-inflammatory cytokines and alleviate the normal tissue inflammatory damage [5,6]. Increasing the amount of probiotics or dietary fiber in the diet could significantly improve inflammatory enteritis symptoms, but the exact mechanism remains unclear [7,8].

Fluorouracil (5-FU) is a commonly used tumor chemotherapy drug that inhibits tumor cell proliferation by interfering with tumor cell DNA synthesis [9]. It is widely used in the treatment of a variety of cancers. There are, however, 40–80% of patients who experience intestinal mucositis-like symptoms during treatment, including indigestion, diarrhea, and dehydration, which in severe cases can prevent chemotherapy from continuing [10]. Previous studies had shown that oral probiotics/prebiotics, such as lactobacillus or dietary fiber, could significantly reduce intestinal side effects and improve patients’ tolerance to chemotherapy drugs, but the exact mechanism is still unclear [11,12]. The release of pro-inflammatory cytokines by activated macrophages in the mucosa is an important cause of chemotherapeutic intestinal mucositis [13]. It will be helpful to understand how intestinal microflora metabolites inhibit macrophage inflammatory response to provide a new approach to prevent and treat chemotherapeutic intestinal mucositis. Our previous studies had shown that the three main SCFAs could reduce the 5-FU-induced inflammatory response in human mononuclear macrophage (THP−1) cells and down-regulate the activation of the NLR family, pyrin domain-containing protein 3 (NLRP3) inflammasome in THP−1 cells, along with production of important pro-inflammatory factors and reactive oxygen species (ROS) [14].

The glycerolphospholipid and sphingolipid pathways are important lipid metabolism pathways in the body and are involved in many important signal transduction processes, such as regulating cell growth, differentiation, and aging [15,16,17]. In these metabolic pathways, ceramide plays a key role in apoptosis, inflammation, and stress responses [18]. In particular, ceramide is significantly up-regulated in 5-FU-resistant cells [19]. Furthermore, the excessive accumulation of ceramide induces the activation of the NLRP3 inflammasome [20]. NLRP3 inflammasomes are composed of apoptosis-associated granule-like protein (ASC), assembly linked NLRP3, and pro-Caspase−1, which play an important role in initiating inflammation [21]. Our previous study found that activated NLRP3 inflammasomes increase the release of pro-inflammatory factors, which greatly contributes to the progression of the inflammatory response induced by 5-FU [14].

Studies have shown that microbial products, such as SCFAs and sphingolipids, act on inflammatory bowel disease [22]. However, few studies have explored the mechanism of action of SCFAs in the treatment of 5-FU-induced intestinal mucosal inflammation from the perspective of the glycerolphospholipid and sphingolipid pathway. Therefore, we hypothesized that SCFAs prevent 5-FU-induced intestinal mucosal inflammation by inhibiting the activation of NLRP3 inflammasomes through the regulation of the glycerolphospholipid and sphingolipid pathways. In this study, we investigated the mechanisms by which the three important SCFAs inhibit 5-FU-induced THP−1 cellular inflammation using untargeted metabolomics and explored the feasibility of intestinal flora metabolites to alleviate 5-FU-induced intestinal mucosal inflammation.

## 2. Results

### 2.1. The ROS Levels in THP-1 Cells

The expression of ROS in THP−1 cells was determined by flow cytometry (Figure 1). Compared with the NC group, ROS expression in THP−1 cells was significantly increased in the 5-FU group (*p* < 0.05). Compared with the 5-FU group, the expression of ROS in the NaAc, NaPc, and NaB groups was decreased significantly (*p* < 0.05).

### 2.2. The Nuclear and Cytoplasmic Distribution of NF−κB p65 in THP−1 Cells

The expressions of NF−κB p65 in the nucleus and cytoplasm of THP−1 cells were determined by Western blotting (Figure 2). Compared with the NC group, after treatment with 2.5 mmol/L of 5-FU, NF−κB p65 expression in the nucleus was significantly up-regulated, and NF−κB p65 expression in the cytoplasm was significantly down-regulated in the 5-FU group (*p* < 0.05). Compared with the NC group, NF−κB p65 expressions in the nucleus of the 5-FU group and the NaAc and NaB groups were significantly increased (*p* < 0.05), and NF−κB p65 expressions in the cytoplasm of the 5-FU, NaAc, NaPc, and NaB groups were significantly decreased (*p* < 0.05). In comparison with the 5-FU group, the expressions of NF−κB p65 in the nucleus of the NaAc, NaPc, and NaB groups decreased significantly. However, NF−κB p65 expressions in the cytoplasm were significantly upregulated (*p* < 0.05).

### 2.3. The Expressions of Pro-Inflammatory Factors in THP-1 Cells

In the qRT-PCR assay, compared with the NC group, the mRNA expressions of NLRP3, Caspase−1, IL−1β, and IL−6 in the 5-FU group were significantly higher (Figure 3). Compared with the 5-FU group, the mRNA expressions of NLRP3, Caspase−1, IL−1β, and IL−6 in the NaAc, NaPc, and NaB groups decreased significantly (*p* < 0.05). Compared with the 5-FU group, the mRNA expression of IL−10 in the NaAc, NaPc, and NaB groups increased significantly (*p* < 0.05). In the ELISA assay, compared with the NC group, the level of IL−1β in the 5-FU group was significantly increased, and the level of IL−10 was significantly decreased (*p* < 0.05) (Figure 3). Compared with the 5-FU group, the level of IL−1β in the NaAc, NaPc, and NaB groups decreased significantly, and the level of IL−10 in the NaAc, NaPc, and NaB groups increased significantly (*p* < 0.05).

### 2.4. NLRP3 Knockdown on 5-FU-Induced IL-1β Expression

When transfected with NLRP3-siRNAs for 24 h, NLRP3 expression was significantly decreased compared to the NC group in the siRNA knockdown. While in the siRNA control, NLRP3 expression was significantly increased (*p* < 0.05). IL−1β levels in the supernatant of the cell culture were used to evaluate the inflammatory response of THP−1 cells. In the siRNA control, IL−1β levels in the 5-FU group were significantly increased compared with the NC group (*p* < 0.05). In the siRNA knockdown, there was no significant difference in the IL−1β level between the NC group and the 5-FU group (Figure 4).

### 2.5. Regulation of Metabolomics in THP-1 Cells

In this study, UPLC−MS/MS was used to comprehensively analyze metabolic profiling and metabolites among the NC, 5-FU, NaA, NaP, and NaB groups. Two hundred and forty−seven ions in the positive mode ions were detected in the cell samples. PCA was used for the first time for an unsupervised comprehensive observation of cell samples. Samples from the 5-FU group showed a deviation from the NC group in terms of metabolite profiles, suggesting that the endogenous metabolites in the cells changed significantly after model making (Figure 5). Compared with the 5-FU group, the NaAc, NaPc, and NaB groups were closer to the NC group in the principal component direction. Simultaneously, the separating trends among the NaA, NaP, NaB, and 5-FU groups were clear.

We then performed OPLS−DA to further analyze and characterize group differences. In the graphs of the OPLS−DA scores, there was a clear separation between the NC, 5-FU, NaAc, NaPc, and NaB groups (Figure 6). In the OPLS−DA spots, five clusters separated well and clustered distinctly in positive ion modes, reflecting that the group differences were more prominent than the individual ones. The evaluation parameters of the OPLS−DA model were R^2^X = 0.729, R^2^Y = 0.988, and Q^2^ = 0.976 (positive ion mode), where R^2^ represents the interpretability of the variables, Q^2^ represents the predictability of the model, and the closer these two are to 1 represents a better OPLS−DA model explanation of the group differences. Cells after the administration of the NaAc, NaPc, and NaB groups were significantly distinct from the 5-FU group, indicating that these SCFAs affected THP−1 cell metabolic profile, and the metabolite disorders were improved, skewing them toward normal cells.

As a supervised multivariate analysis method, OPLS−DA showed the distinct separation of the NC and 5-FU groups in the positive ionic modes. The ions with VIP > 1.0 and *p* < 0.05 were significant differential metabolites in the present study. Twenty-four cell metabolites with significantly altered expression were screened between the 5-FU and NC cohorts. On applying the OPLS−DA model to evaluate the changes among the NaAc, NaPc, NaB, and 5-FU groups, the metabolic profile between the 5-FU group and the NaAc, NaPc, and NaB groups were shown to be completely separated, and 20 significantly altered cellular metabolites were screened in the NaAc, NaPc, NaB, and 5-FU groups (Table 1).

Heat maps show trends in overall changes in common characteristics and metabolite levels among different populations (Figure 7A). Metabolites with similar abundance trends were placed closer together. According to the heat map results, the three major metabolites treated with SCFAs were closely clustered and separated from the 5-FU group. Next, we applied the online KEGG database to pathway enrichment analysis, exploring the most relevant pathways as well as potential mechanisms. Variations in the 20 disturbed metabolites suggested that the differences between the 5-FU group and the NaAc, NaPc, and NaB groups were related to multiple metabolic pathways, including D-glutamine and D-glutamate metabolism, glycerophospholipid metabolism, nitrogen metabolism, arginine metabolism, butanoate metabolism, histidine metabolism, sphingolipid metabolism, galactose metabolism, and glutathione metabolism (Figure 7B). The callback trends of disrupted metabolites suggested that the anti-inflammation mechanism of the three major SCFAs was related to the metabolic pathways.

## 3. Discussion

It is known that 5-FU not only interferes with DNA synthesis in tumor cells, but also in normal cells [23]. In turn, this leads to a large amount of ROS being produced and a large amount of double-stranded DNA (dsDNA) being released, which act as damage-associated molecular patterns (DAMPs) [24,25]. dsDNA plays a significant role in inflammatory and autoimmune diseases [26]. The NLRP3 family is the most important member of the inflammasome family, and it can be activated by bacterial toxins, ATP, ROS, other pathogens, and dangerous signaling molecules [27]. The NLRP3 inflammasome is an important factor in anti-infective immunity and the induction of inflammatory diseases [28]. Immune cells and epithelial cells of the intestinal mucosa are not highly expressed when there are no infections or intestinal inflammation [29,30]. Upon activation, NLRP3 expression is up-regulated in intestinal immune cells. Then, the assembly of activated protease Caspase-1 promotes the cleavage and maturation of IL−1β and IL−18 precursors, triggering a severe inflammatory response [31,32]. In this study, 5-FU up-regulated ROS levels in THP-1 cells and then stimulated the entry of NF−κB65 into the nucleus, activating the NF−κB65 signaling pathway and activating NLRP3 inflammasome expression, which ultimately leads to an increased expression of various pro-inflammatory factors.

SCFAs are the major product of anaerobic fermentation, which is closely related to the regulation of the intestinal immune system [33,34]. SCFA concentrations in the human intestinal tract range from 70 to 140 mmol/L and are rapidly absorbed by the blood, which mainly includes NaAc, NaPc, and NaB [35]. Aside from producing energy for epithelial cells, SCFAs can regulate flora composition in the human intestinal tract through fermentation, reduce the growth of harmful bacteria, and prevent intestinal dysfunction [36,37]. Furthermore, SCFAs may inhibit the release of pro-inflammatory factors from immune cells, thereby inhibiting excessive intestinal inflammation and reducing intestinal mucosal injury in patients with colitis [38,39]. According to previous studies, all three main SCFAs reduce immune cell release of pro-inflammatory factors, inhibit inflammation, and promote the expression of anti-inflammation factors [40]. Moreover, NaAc and NaB can restrict the activation of the NF−κB pathway, which can also inhibit inflammation [40], and the results we obtained support this conclusion. According to our results, the three main SCFAs could up-regulate the expression levels of the anti-inflammatory cytokine IL−10. There is evidence that these IL−10 can significantly suppress the expression of pro-inflammatory factors, including ROS and NF−κB pathway activation, and NLRP3 inflammasome activity [38].

In summary, our metabolomics results showed that 5-FU-induced THP−1 cell metabolites were disordered, and the three SCFAs can effectively regulate these differential metabolites across multiple metabolic routes. Several different metabolites are described below in terms of their biological functions. Sphingolipids, including ceramide, sphingosine-1-phosphate (S1P), sphingosine, sphingolipids, and glycosphingolipids, serve as basic components of organelles and membranes [41]. Sphingolipid molecules have multiple biological functions and serve as essential components of organelles and membranes [42]. For example, ceramide and sphingosine can effectively inhibit the growth of myeloma cells, rhabdomyosarcoma cells, renal tubular cells, and hippocampal nerve cells and induce their apoptosis [43,44,45], whereas S1P can inhibit apoptotic macrophages, granule cells, and other cells [46,47]. This study found that the three SCFAs’ intervention in THP−1 cells reduces sphingosine generation. Sphingolipid metabolism is likely to be affected by the three SCFAs, thereby promoting cell growth and reducing apoptosis.

Amino acids are important nutrients for immune cells and are the basic building blocks of the human immune system. Multiple amino acids such as glutamate, arginine, tryptophan, leucine, methionine, and cysteine exert activation, differentiation, and function in T cells [48]. The enhancement of macrophage function by glutamine mainly promotes antigen presentation, phagocytosis, and cytokine secretion [49], and glutamine supplementation reduces pro-inflammatory IL−6 and IL−8 production in lymphocytes and epithelial cells and enhances the expression of anti-inflammatory IL−10 [50]. In the present study, NaAc can significantly up-regulate glutamate production and promote amino acid metabolism. This suggests that the mechanism by which NaAc ameliorates 5-FU-induced inflammation might be related to the regulation of amino acid metabolism. 

In signal transduction, glycerophospholipids function as precursors for lipid mediators. In the cell membrane, phosphatidylcholine is the main glycerophospholipid. Phosphatidylcholine, a major constituent of the lipid bilayer structure of the cell membrane, plays a significant role in cell membrane fusion, pinocytosis, and membrane function [51]. It has been found that SCFAs could regulate the metabolism of phosphatidylcholine by the gut microbiota [52]. Acetate and propionate are mainly metabolized in the liver, and the substrates for gluconeogenesis serve as an energy source and are involved in fatty acid synthesis [53]. Tsutsumi et al.’s results confirmed that intestinal colonization of SCFA-producing bacteria, such as Bacteroidetes, could promote the synthesis of long-chain fatty acids (C16 and C18) and their precursors of phosphatidylcholine in the liver [54]. In this study, the three main SCFAs on THP−1 cells could increase phosphatidylcholine production, indicating that the mechanisms of anti-inflammation induced by 5-FU may be related to the regulation of glycerolphospholipid metabolism.

## 4. Materials and Methods

### 4.1. Cell Culture and Processing

Human mononuclear macrophage (THP−1) cells were used in this experiment and cultured and subcultured according to conventional methods [55]. The cells were cultured in RPMI 1640 medium (Biological Industries, Kibbutz, Israel) containing 10% fetal bovine serum (Biological Industries, Kibbutz, Israel) at 37 °C and 5% CO_2_. The cells, in the logarithmic growth phase, were inoculated in a 6-well plate with 1 × 10^6^ cells per well. A normal control group (NC), a 5-FU group (5-FU), a 5-FU + NaAc, a 5-FU + NaPc, and a 5-FU + NaB were set up in the experiment. In this experiment, THP−1 cells were induced to macrophage formation with a concentration of 100 ng/mL phorbol-12-myriestate-13-acetate (PMA). The cells in the NC group did not undergo another treatment. The cells in the NaAc group, NaPc group, and NaB group were pre-treated with 100 μmol/L NaAc, NaPc, and NaB for 24 h, respectively. Then, they were treated with 2.5 mmol/L 5-FU for 24 h. The cells in the 5-FU group, NaAc group, NaPc group, and NaB group were only treated with 2.5 mmol/L 5-FU for 24 h. 

### 4.2. ROS Detection

The 2-dichlorofluorescein yellow diacetate (DCFH−DA) probe kit (Beyotime Biotechnology, China) was used for the detection of intracellular ROS concentration [56]. The serum−free medium diluted DCFH−DA at the ratio of 1:1000 to a final concentration of 10 μmol/L. The cells were resuspended in diluted DCFH−DA probe solution, incubated in an incubator at 37 °C for 20 min, and mixed upside down every 5 min so that the probe was in complete contact with the cells. The serum-free cell culture medium was washed 3 times to remove the DCFH−DA probe that did not enter the cell, and then, the cells were resuspended with aseptic phosphate buffer solution for detection, following the kit instruction. The fluorescence intensity of DCFH−DA in cells was detected by flow cytometry (Accuri C6 Plus, BD, New Jersey, NJ, USA). All the experiments were replicated 3 times.

### 4.3. Western Blotting Analysis

The process of Western blotting assessment followed Yue’s report [14]. The primary antibodies were diluted 1:1000 in TBST buffer and incubated overnight at 4 °C. The ECL chemiluminescence kit was applied (Millipore, MA, USA), and the results were analyzed by Image J software. Total cellular protein expression and cytoplasmic protein analysis were performed using β−actin as a reference. The expression of nuclear protein analysis was performed using Histone H1 as a reference. The separation of nuclear protein and cytoplasmic protein was as follows. An amount of 200 μL of cytoplasmic protein extraction reagent A was added into the collected cells, and the cells were thoroughly dispersed by shaking violently for 5 s. The cells were placed on ice for 10 min. Then, cytoplasmic protein extraction reagent B 10 μL was added, and the reagent was shaken violently for 5 s and placed on ice for 1 min. After centrifugation at 16,000× *g* at 4 °C for 5 min, the supernatant was the extracted cytoplasmic protein, and the precipitation was the nuclear protein. All the experiments were replicated 3 times.

### 4.4. qRT−PCR Analysis

The total RNA isolation, reverse transcription, and quantitative real-time reverse transcription–polymerase chain reaction (qRT−PCR) were performed according to the standard protocol described by Feng et al. [57]. Total RNA was purified from THP-1 cells using Trizol method, and cDNA was synthesized by reverse transcription with oligo (dT)n as a primer. mRNA expression was determined by qRT−PCR. The qRT−PCR volume was 20 μL, including 10 μL of SYBR Green Master premixture, 0.4 μL of upstream primer, 0.4 μL of downstream primer (10 μmol/L), and 2 μL of cDNA template. The reaction procedure included 95 °C pre-denaturation for 5 min, 95 °C denaturation for 3 s, 58 °C annealing for 20 s, and 72 °C extension for 30 s. The whole qPCR reactions were 40 cycles. The GAPDH gene was the reference, and the relative expression of mRNA was calculated by the 2^−ΔΔCT^ method. All the experiments were replicated 3 times. The primer sequences are listed in Appendix A.

### 4.5. ELISA Analysis

The concentration of IL−1β in the supernatant of the cell culture medium was detected by ELISA assay (MEIMIAN, Yangcheng, China) in accordance with the manufacturer’s instructions. 

### 4.6. NLRP3 Knockdown

The NLRP3-siRNA product was purchased from GENEWIZ company, Suzhou, China. The transfection operation was performed when the cell growth and fusion reached 60% in the cell culture plate. According to the instructions for ExFect 2000 Transfection Reagent (Vazyme, Suzhou, China), 25 μL serum-free medium and 3 μL transfection reagent were added into 1.5 mL sterile centrifuge tube, blown and mixed, and left at room temperature for 5 min as the siRNA knockdown group. Next, 25 μL serum-free medium and 3 μL siRNA-NLRP3 were added to a new 1.5 mL aseptic centrifuge tube and then left for 5 min at room temperature as the siRNA knockdown group. Then, the liquid in the two centrifuge tubes was fully mixed and left for 10 min at room temperature before transfection. The ExFect 2000/siRNA-NLRP3 mixture was added into the cell culture dish drop by drop, and the cells were cultured for 24 h to knockdown NLRP3 expression. This experiment included a siRNA control group and siRNA knockdown group, and each group included the NC group and the 5-FU group. The cells in the two 5-FU groups were treated with 2.5 mmol/L 5-FU for 24 h. Then, the cells were collected and determined by Western blotting analysis. The IL−1β level in the supernatant of cell culture medium was used as an indicator to evaluate the inflammatory response. The level of IL−1β was determined by ELISA assay. 

### 4.7. Metabolic Characterization of Cell Sample Preparation 

Cell metabolomics analysis samples were prepared according to Liu’s method [58]. Metabolomics detection was performed by the Testing Center of Yangzhou University using chromatography–mass spectrometry platform. Chromatographic separation of the metabolites was performed on a Waters UPLC system equipped with an ACQUITY UPLC^®^ BEH C18 (100 mm × 2.1 mm i.d., 1.7 µm; Waters, Milford, MA, USA). The mobile phases were composed of 0.1% formic acid in water (A) and acetonitrile (B) with a flow rate of 0.3 mL/min. The solvent gradient varied according to 0–3 min, 99–80% A; 3–4 min, 80–30% A; 4–20 min, 30–25% A; 20–21 min, 25–1% A; followed by 4 min for equilibrating the systems. A sample injection volume of 2 μL was used and column temperature was maintained at 40 °C. All these samples were kept at 4 °C during the analysis period. The mass spectrometric data were collected using a Waters ACQUITY UPLC I-Class Plus/VION IMS QTof Mass Spectrometer. The parameter settings for the ESI ionization mode were set as follows: analyzer mode, sensitivity; capillary, 0.8 kV; source temperature, 120 °C; desolvation temperature, 550 °C (positive) and 450 °C (negative); cone gas flow rate, 50 L/h; desolvation gas flow rate, 800 L/h; and collision energy contained low energy 6 V and energy ramp 10 to 40 eV. Data were acquired with mass spectrometry (MS^E^) technique. The full MS resolution was 50,000. The mass range for detection is 50–1000 *m*/*z*.

### 4.8. Data Processing and Multivariate Analysis

Raw data were imported into Progenesis QI 2.3 (Nonlinear Dynamics, Waters, Milford, MA, USA) for peak detection and alignment following UPLC-MS/MS analyses. Preprocessing generated a data matrix consisting of the retention time (RT), mass-to-charge ratio (*m*/*z*) values, and peak intensity. Metabolic traits detected by at least 80% were retained in any sample set. After filtration, the lowest metabolite values for a specific sample were estimated. Metabolite levels fell below the quantitative threshold, and metabolic characteristics were normalized according to the total amount. We used the internal standard for QC data (reproducibility), metabolic characteristics, and discarded the relative standard deviation (RSD) of the >30% QC. After normalization procedures and imputation, statistical analysis was performed on log-transformed data to identify significant differences in metabolite levels between comparable groups. The accurate mass identified the mass spectra of these metabolic features, MS/MS fragments spectra, and isotope ratio difference by searching reliable biochemical databases, such as the Human Metabolome Database (HMDB) (http://www.hmdb.ca/, accessed on 6 October 2022) and Lipid Maps (https://www.lipidmaps.org/, accessed on 6 October 2022). The mass tolerance between the measured *m*/*z* values and the exact mass of the components of interest was ±10 ppm. For metabolites with MS/MS confirmation, only those with MS/MS fragment scores greater than 30 were considered confidently identified as metabolites. In order to obtain an overview of the metabolic data, principal component analysis (PCA) using an unsupervised method was applied through the visualization of general clustering, trends, or outliers. Orthogonal partial least squares discriminant analysis (OPLS-DA) was used to build a model for identifying variables. Significantly different metabolites were selected based on the variable weight value (VIP) obtained from OPLS-DA model and Student’s *t* test *p*-value. Metabolites with VIP > 1.0 and *p* < 0.05 were considered potential biomarkers for the analysis.

### 4.9. Statistical Analysis

The results were presented as mean ± standard errors of the mean (SEM). Image J Software and GraphPad Prism 8.0 analysis software were used for image analysis and data processing after protein development, and data were expressed as mean ± standard deviation. SPSS 22.0 software was used for performing the t-test and one-way ANOVA. A value of *p* < 0.05 was considered to be statistically significant.

## 5. Conclusions

In conclusion, 5-FU can increase the expression of ROS, activate the NF-κB/NLRP3 pathway, and, finally, induce and aggravate cellular inflammation; the three main SCFAs (NaAc, NaPc, and NaB) can inhibit cellular inflammation and play a therapeutic role through inhibiting NF-κB/NLRP3 signaling via regulating glycerolphospholipid and sphingolipid metabolism in THP-1 cells (Figure 8). In addition, although the three main SCFAs can inhibit the inflammation of macrophages, the inhibitory effect of the mixtures of the three main SCFAs in different proportions remains to be further studied.

## Figures and Tables

**Figure 1 molecules-28-00494-f001:**
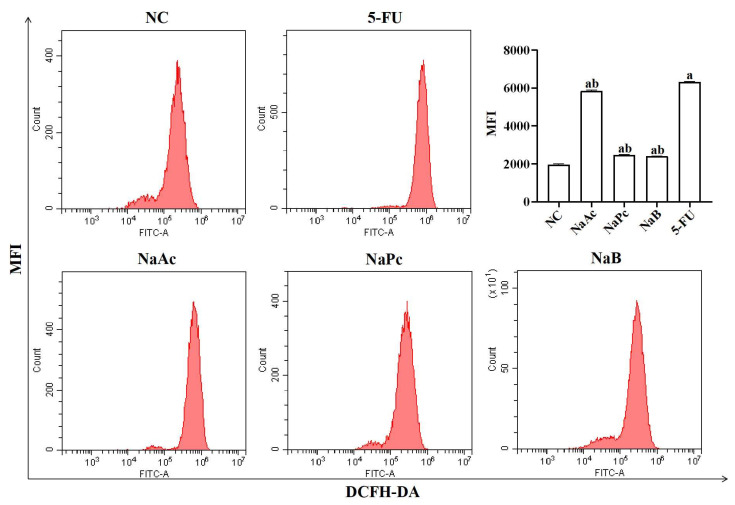
Detection of intracellular ROS expression in THP-1 cells by flow cytometry (*n* = 3/group). The three main SCFAs (NaAc, NaPc, and NaB) can significantly inhibit the ROS expression induced by 2.5 mmol/L of 5-FU. Each experiment was repeated at least three times. Data are expressed as the mean ± SEM. a—*p* < 0.05, vs. the NC group; b—*p* < 0.05, vs. the 5-FU group.

**Figure 2 molecules-28-00494-f002:**
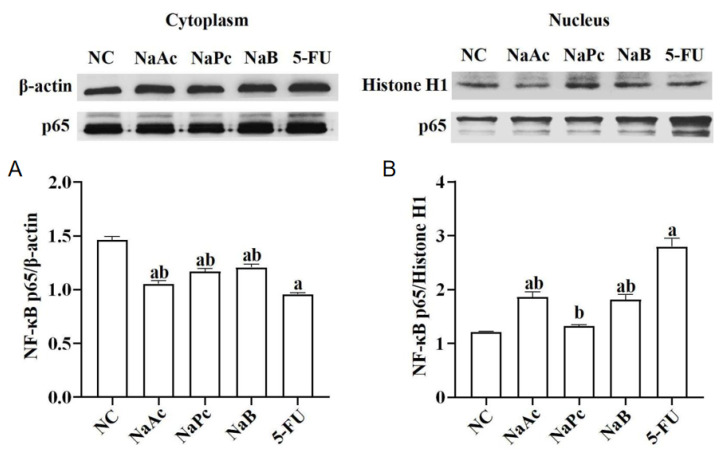
Effects of the SCFAs on NF−κB p65 expression (*n* = 3/group). The cells were pretreated with 100 μmol/L of SCFAs (NaAc, NaPc, and NaB) and then stimulated with 5-FU (2.5 mmol/L). The expression of NF−κB p65 in cell nuclear (**A**) and cytoplasmic (**B**) lysates were determined by Western blotting. Each experiment was repeated at least three times. Data are expressed as the mean ± SEM. a—*p* < 0.05, vs. the NC group; b—*p* < 0.05, vs. the 5-FU group.

**Figure 3 molecules-28-00494-f003:**
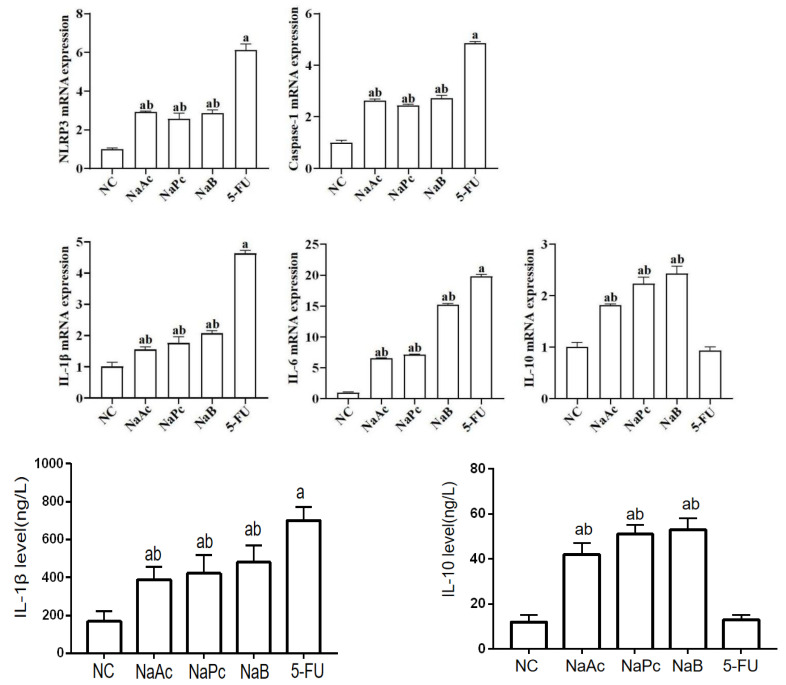
Effects of the SCFAs on inflammatory mediator expression by qRT-PCR and ELISA assay (*n* = 3/group). The cells were pretreated with 100 μmol/L of SCFAs (NaAc, NaPc, and NaB) and then stimulated with 5-FU (2.5 mmol/L). The mRNA expressions of NLRP3, Caspase−1, IL−6, and IL−10 were determined by qRT-PCR, and the contents of IL−1β and IL−10 were determined by ELISA. Each experiment was repeated at least three times. Data are expressed as the mean ± SEM. a—*p* < 0.05, vs. the NC group; b—*p* < 0.05, vs. the 5-FU group.

**Figure 4 molecules-28-00494-f004:**
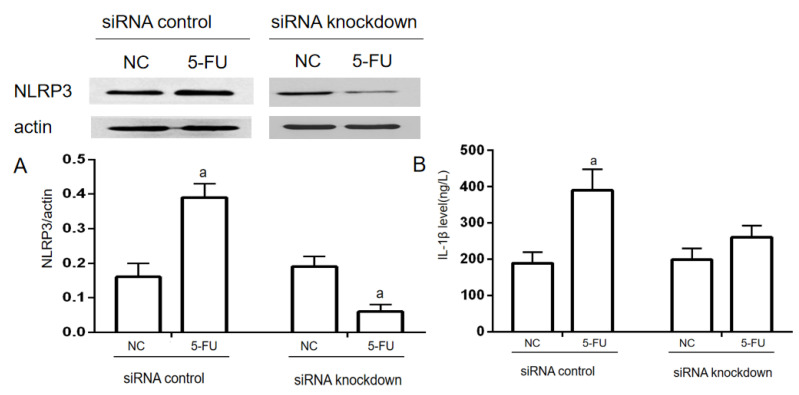
NLRP3 knockdown on 5-FU-induced IL−1β expression in THP−1 cells (*n* = 3/group). NLRP3-siRNA was used for NLRP3 knockdown in THP−1 cells. The siRNA-NLRP3 and ExFect 2000 Transfection Reagent mixture was added into the culture medium for 24 h. Then, the final concentration of 2.5 mmol/L 5-FU was added into the medium to induce the inflammatory response. (**A**) The expression of NLRP3 was determined by Western blotting assay. (**B**) The level of IL−1β in the culture medium was determined by ELISA assay. Each experiment was repeated at least three times. Data are expressed as the mean ± SEM. a—*p* < 0.05, vs. the NC group.

**Figure 5 molecules-28-00494-f005:**
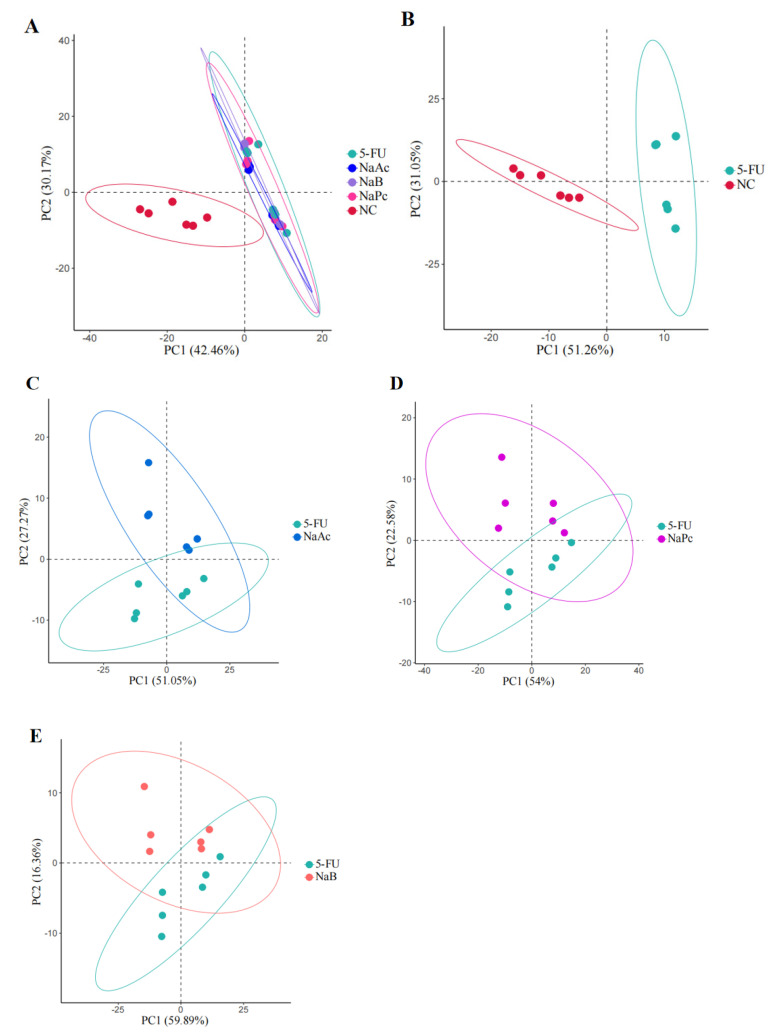
PCA score plot in the positive ion mode analysis for the effects of SCFAs on cellular metabolites (*n* = 6/group). (**A**): PCA score plot analysis in all groups; (**B**): PCA score plot analysis for the NC and 5-FU group; (**C**): PCA score plot analysis for the NaAc and 5-FU group; (**D**): PCA score plot analysis for the NaPc and 5-FU group; and (**E**): PCA score plot analysis for the NaB and 5-FU group.

**Figure 6 molecules-28-00494-f006:**
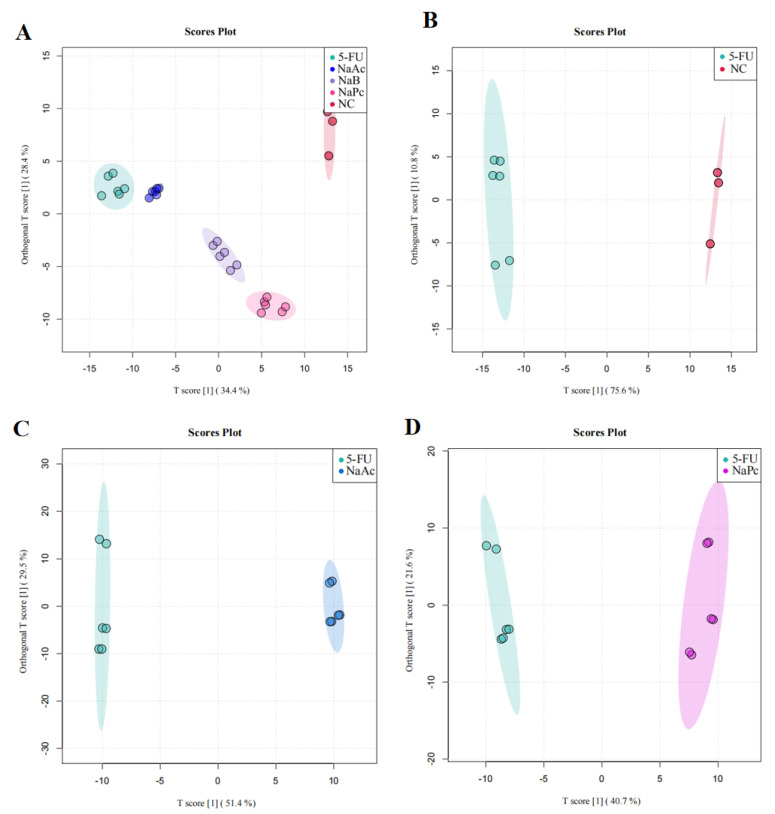
OPLS−DA score plot in the positive ion mode analysis for the effects of SCFAs on cellular metabolites (*n* = 6/group). (**A**): OPLS−DA score plot analysis in all groups; (**B**): OPLS−DA score plot analysis for the NC and 5-FU group; (**C**): OPLS−DA score plot analysis for the NaAc and 5-FU group; (**D**): OPLS−DA score plot analysis for the NaPc and 5-FU group; and (**E**): OPLS−DA score plot analysis for the NaB and 5-FU group.

**Figure 7 molecules-28-00494-f007:**
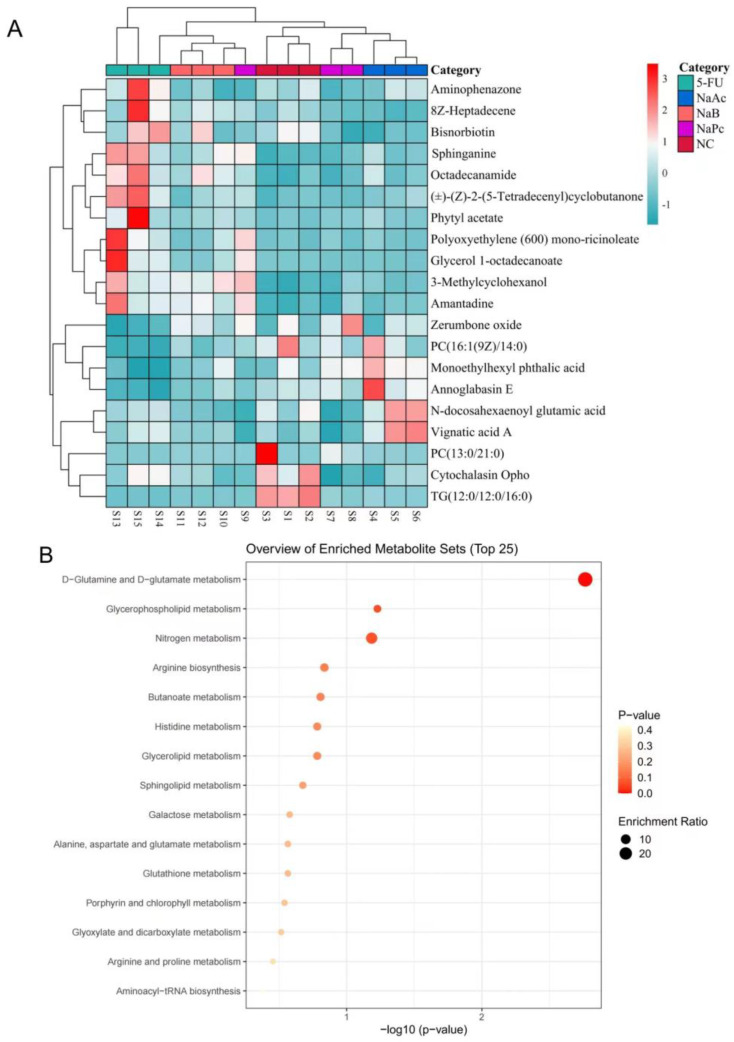
The heat maps of differential metabolites and the main metabolic pathway impact analysis. (**A**): heat maps of differential metabolites from cells. Rows: metabolites; Columns: samples. On the top is the cluster of samples, and on the left is the cluster of metabolites. Red means the metabolites were expressed at a higher level, and blue means the metabolites were expressed at a lower level. (**B**): the main metabolic pathway impact analysis. The smaller the *p*-value, the more significant the enrichment. The magnitude of the enrichment factor indicates the reliability of significance.

**Figure 8 molecules-28-00494-f008:**
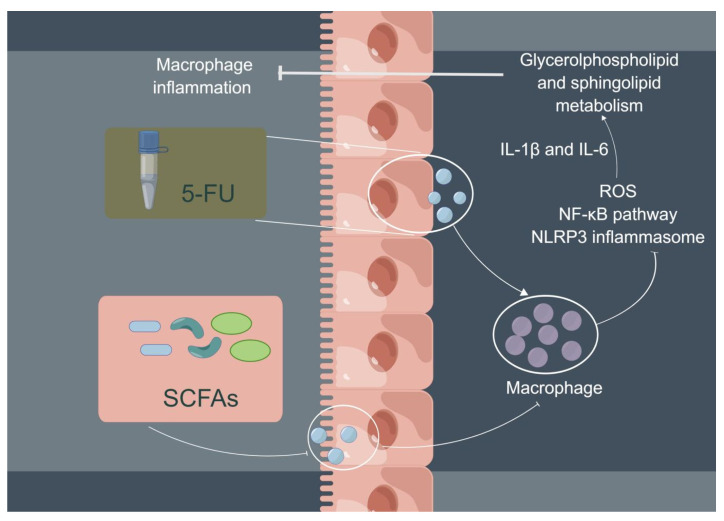
Potential mechanism of intestinal SCFAs inhibiting 5-FU-induced macrophage inflammation. 5-FU can activate intestinal mucosal macrophages, and activated macrophages secrete a large number of pro-inflammatory cytokines to cause intestinal mucosal damage (chemotherapy intestinal mucositis). SCFAs are beneficial metabolites produced by bacteria in the gut. Three main SCFAs (acetate, propionate, and butyrate) inhibited 5-FU-induced THP-1 cell inflammation and reduced pro-inflammatory cytokine expression (IL-1β and IL-6) via inhibiting NF-κB signaling pathway and NLRP3 inflammasome activation by regulating glycerolphospholipid and sphingolipid metabolism.

**Table 1 molecules-28-00494-t001:** Cellular metabolites of THP-1 cells treated with SCFAs.

No.	Metabolite	Formula	Library ID	RT/min	M/Z	NaAc	NaPc	NaB
1	Sphinganine	C18H39NO2	HMDB00269	5.02	302.31	↓	↓	↓
2	Polyoxyethylene (600) mono-ricinoleate	C21H40O3	HMDB32476	21.65	703.57	↓	↓	↓
3	(±)-(Z)-2-(5-Tetradecenyl)cyclobutanone	C18H32O	HMDB37543	12.7	282.28	↓	↓	↓
4	Cytochalasin Opho	C28H37NO4	HMDB35366	5.66	452.27	↓	↓	↓
5	N-docosahexaenoyl glutamic acid	C27H39NO5	LMFA08020089	5.96	480.27	↑	↓	↓
6	Phytyl acetate	C22H42O2	HMDB32470	5.52	356.35	↓	↓	↓
7	Amantadine	C10H17N	HMDB15051	4.83	320.31	↓	↓	↓
8	Octadecanamide	C18H37NO	HMDB34146	5.03	284.29	↓	↓	↓
9	8Z-Heptadecene	C17H34	LMFA11000102	5.23	256.3	↓	↓	↓
10	Glycerol 1-octadecanoate	C21H42O4	HMDB31075	21.64	359.31	↓	↓	↓
11	Phosphatidylcholine (13:0/21:0)	C42H84NO8P	LMGP01010466	21.75	784.58	↑	↑	↑
12	Phosphatidylcholine (16:1(9Z)/14:0)	C38H74NO8P	LMGP01011475	21.77	726.5	↑	↑	↑
13	Annoglabasin E	C20H32O3	HMDB36262	9.71	658.51	↑	↑	↑
14	Aminophenazone	C13H17N3O	HMDB15493	6.55	480.31	↓	↓	↓
15	Zerumbone oxide	C15H22O2	HMDB36466	6.36	235.16	↑	↑	↑
16	Bisnorbiotin	C8H12N2O3S	HMDB04821	6.53	234.09	↓	↓	↓
17	Monoethylhexyl phthalic acid	C16H22O4	HMDB13248	7.21	579.29	↑	↑	↑
18	Vignatic acid A	C30H39N3O7	HMDB33599	6.11	554.28	↑	↓	↓
19	TG (12:0/12:0/16:0)	C43H82O6	LMGL03012632	3.56	712.64	↑	↑	↑
20	3-Methylcyclohexanol	C7H14O	HMDB31538	4.64	246.24	↓	↓	↓

Note: ↑ indicates increase, ↓ indicates decrease, vs. 5-FU group.

## Data Availability

The data presented in this study are available on request from the corresponding authors.

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
