# Peer review of "Short-Chain Fatty Acids Attenuate 5-Fluorouracil-Induced THP-1 Cell Inflammation through Inhibiting NF-κB/NLRP3 Signaling via Glycerolphospholipid and Sphingolipid Metabolism"

_molecules, 2023, doi:10.3390/molecules28020494_

Round 1
Reviewer 1 Report
The authors investigated the anti-inflammatory effects of the three important short chain fatty acids (SCFAs), including sodium acetate (NaAc), sodium propionate (NaPc), and sodium butyrate (NaB), on human mononuclear macrophage-derived THP-1 cells induced by 5-FU. The results showed that the three SCFAs inhibited the pro-inflammatory factors expressions including IL-1β, and IL-6 when treated with 5-FU. However, there are a few issues that need to be considered and required correction:
Page 1; Line 27-28: It would be great if the statistically significant values of inhibition for the pro-inflammatory factors expressions including IL-1β, and IL-6 when treated with 5-FU should be used in the abstract.
Page 1; Line 29-30: It would be great if the statistically significant values of inhibition for ROS expression and NF-κB activity of 5-FU-treated THP-1 cell by the three SCFAs pre-incubated should be used in the abstract.
Page 3; Line 102-103: The results in section 2.2 should be shown by concentration such as which treatment concentration with 5-FU, NF-κB p65 expression in the nucleus was significantly up-regulated, and significantly down-regulated NF-κB p65 expression in the cytoplasm.
This pattern should be followed throughout all the sections of the results in the manuscript.
Page 10; Line 288: The methods for culturing mononuclear macrophage (THP-1) cells should be given in brief even following the previous methods.
Page 10; Line 289: Please confirm the number of cells incubated in a 6-well plate with 1 × 106 cells or 1 x 106 in each well.
Author Response
Reviewer 1#
The authors investigated the anti-inflammatory effects of the three important short chain fatty acids (SCFAs), including sodium acetate (NaAc), sodium propionate (NaPc), and sodium butyrate (NaB), on human mononuclear macrophage-derived THP-1 cells induced by 5-FU. The results showed that the three SCFAs inhibited the pro-inflammatory factors expressions including IL-1β, and IL-6 when treated with 5-FU. However, there are a few issues that need to be considered and required correction:
Dear Reviewer,
Thank you for your rapid response on our manuscript. The constructive criticism of you was much appreciated and we revised our manuscript accordingly. The suggestions were accepted, more information and details were included in the text, and the manuscript was revised thoroughly. On the other hand, we rearranged the manuscript to improve the quality. All the modifications performed in the revised manuscript are highlighted in red. The enclosed document at the bottom of this letter contains a point-to-point reply to you comments.
Page 1; Line 27-28: It would be great if the statistically significant values of inhibition for the pro-inflammatory factors expressions including IL-1β, and IL-6 when treated with 5-FU should be used in the abstract.
Response: Thanks for your comments. We had revised.
Page 1; Line 29-30: It would be great if the statistically significant values of inhibition for ROS expression and NF-κB activity of 5-FU-treated THP-1 cell by the three SCFAs pre-incubated should be used in the abstract.
Response: Thanks for your comments. We had revised.
Page 3; Line 102-103: The results in section 2.2 should be shown by concentration such as which treatment concentration with 5-FU, NF-κB p65 expression in the nucleus was significantly up-regulated, and significantly down-regulated NF-κB p65 expression in the cytoplasm. This pattern should be followed throughout all the sections of the results in the manuscript.
Response:In this study, 5-FU with a final concentration of 2.5 mmol/L was used to induce an inflammatory response in THP-1 cells. In the results section of this paper, we also added the concentration of 5-FU.
Page 10; Line 288: The methods for culturing mononuclear macrophage (THP-1) cells should be given in brief even following the previous methods.
Response: Thanks for your comments. We had revised.
Page 10; Line 289: Please confirm the number of cells incubated in a 6-well plate with 1 × 106 cells or 1 x 106 in each well
Response: Thanks for your comments. We had revised.

Reviewer 2 Report
Major comment:
According to Table 1, what are the cellular metabolites of untreated THP-1 cells and 5-Fluorouracil-treated cells? This information may provide data on changes in metabolites during 5-Fluorouracil-treatment and how SCFs modulate that.
Explain figure 8 in its legend.
Minor issues:
Line 107-108. Revise the sentence for clarity. And check for the same throughout the manuscript.
Author Response
Reviewer 2#
Major comment:
According to Table 1, what are the cellular metabolites of untreated THP-1 cells and 5-Fluorouracil-treated cells? This information may provide data on changes in metabolites during 5-Fluorouracil-treatment and how SCFs modulate that.
Explain figure 8 in its legend.
Response: Metabolomics detects compounds with molecular weight less than 3kd in cell lysis products, which are mainly derived from the products of cell metabolism. Through the analysis of these metabolites, we can understand what metabolic processes take place in the cell. The explain of Figure 8 was added in the paper.
Line 107-108. Revise the sentence for clarity. And check for the same throughout the manuscript.
Response: Thanks for your comments. We had revised.

Round 2
Reviewer 2 Report
At this point the manuscript is acceptable for publication